# Rapid and Sensitive Glucose Detection Using Recombinant Corn Mn Peroxidase and Advanced Voltammetric Methods

**DOI:** 10.3390/s25195974

**Published:** 2025-09-26

**Authors:** Anahita Izadyar, Ezekiel McCain, Elizabeth E. Hood

**Affiliations:** 1Department of Chemistry and Physics, Arkansas State University, P.O. Box 419, Jonesboro, AR 72467, USA; ezekiel.mccain@smail.astate.edu; 2Affiliation Arkansas Biosciences Institute and College of Agriculture, Arkansas State University, P.O. Box 639, Jonesboro, AR 72467, USA

**Keywords:** electrochemical biosensor, glucose detection, square wave voltammetry (SWV), linear sweep voltammetry (LSV), recombinant corn manganese peroxidase

## Abstract

We present a novel disposable electrochemical biosensor for highly sensitive and selective glucose detection, employing gold-modified screen-printed electrodes combined with square wave (SWV) and linear sweep voltammetry (LSV). The sensor integrates recombinant corn-derived manganese peroxidase with glucose oxidase, bovine serum albumin, and gold nanoparticles to enhance stability and signal transduction. Glucose detection by LSV covered 0.001–6.5 mM (R^2^ = 0.9913; LOD = 0.50 µM), while SWV achieved a broader range of 0.0006–6.5 mM (R^2^ = 0.998; LOD = 0.29 µM). The sensor demonstrated excellent selectivity, showing minimal interference from common electroactive species including caffeine, aspartame, and ascorbic acid, and provided rapid responses, making it ideal for point-of-care and food monitoring applications.

## 1. Introduction

Diabetes mellitus is a prevalent chronic condition affecting millions worldwide and remains a major contributor to morbidity and mortality [1,2]. Effective disease management depends on frequent and accurate monitoring of blood glucose levels, making glucose the most measured analyte in clinical diagnostics. In response to this need, extensive research has been devoted to the development of glucose biosensors. Among them, electrochemical biosensors have emerged as a leading technology due to their high sensitivity, simplicity, low cost, and compatibility with miniaturized, environmentally friendly designs [3,4,5].

A critical factor influencing biosensor performance is the choice of enzyme and its immobilization strategy, as enzyme–substrate interactions drive the sensor’s specificity and sensitivity. Enzymes act as highly selective biocatalysts, facilitating the conversion of the target analyte into detectable electrochemical signals. However, the cost-effective and scalable production of enzymes remains a challenge. Recombinant systems provide an economical means of commercial enzyme production, yet some enzymes particularly oxido-reductases are difficult to express in traditional microbial systems like bacteria and fungi due to issues such as post-translational modifications, solubility, and toxicity to the host [6].

Among alternative production platforms, transgenic plants offer enormous potential, providing unique advantages including scalability, low production cost, and protein stability in storage tissues such as seeds [7,8]. Domesticated crops, particularly maize, are preferred over model plants due to favorable agronomic traits and industrial-scale production capabilities. Maize is especially attractive for recombinant protein expression because of its high yield and the ability to accumulate stable, functional enzymes in the grain. In fact, maize-derived enzymes can reach concentrations of up to 2.8% of dry weight in germ-rich fractions, dramatically reducing cost of goods for large-scale applications.

One such enzyme is manganese peroxidase (MnP), a redox enzyme secreted by lignin-degrading wood-rot fungi [9,10] MnP is part of a group of extracellular enzymes that break down complex aromatic structures and has demonstrated lignin degradation and pulp brightening capabilities [11]. Additionally, MnP shows great promise in analytical and environmental applications, such as detecting glucose, hydrogen peroxide, and emerging pollutants [12,13]. MnP has been successfully expressed in transgenic maize using an embryo-specific promoter, resulting in enzyme accumulation within the corn kernel. The recombinant form, known as plant-produced MnP (PPMP), combines low production cost with high functionality, making it a strong candidate for next-generation biosensor development [14,15,16,17]. To translate these biochemical advantages into practical devices, screen-printed electrodes (SPEs) are widely utilized due to their low cost, design flexibility, and suitability for portable electrochemical sensors [18,19]. These electrodes enable reliable detection of redox-active species and offer advantages such as mechanical stability, high selectivity, and ease of mass production [20,21,22,23]. To improve the sensitivity, stability, and performance of SPE-based biosensors, researchers frequently incorporate conductive nanomaterials and polymers. These materials enhance redox mediation, support enzyme immobilization, and minimize surface fouling [24].

In this context, conductive polymers, especially polyaniline (PANI), have gained attention for their favorable electrochemical properties, cost-effectiveness, and ease of synthesis. PANI exhibits good environmental stability and can be electropolymerized directly onto electrode surfaces, forming conformal coatings that support enzyme attachment [25,26,27,28]. When integrated into nanocomposites with materials like gold nanoparticles (AuNPs), the resulting hybrid systems demonstrate enhanced electrocatalytic activity, stability, and sensitivity [29,30,31,32,33,34].

Electrochemical techniques such as Square Wave Voltammetry (SWV), Linear Sweep Voltammetry (LSV), and Differential Pulse Voltammetry (DPV) provide detailed insights into the behavior of these modified electrodes. SWV is particularly valued for its sensitivity and rapid response, LSV for kinetic analysis of redox systems, and DPV for its precision and low detection limits [35,36,37]. These techniques are indispensable for characterizing biosensor components and evaluating analytical performance [38].

In this study, we report the fabrication of a novel disposable glucose biosensor based on a screen-printed electrode (SPE) modified with polyaniline (PANI), gold nanoparticles (AuNPs), glucose oxidase (GOx), and plant produced MnP (PPMP) derived from recombinant corn. The sensor was constructed using a one-step electropolymerization technique to integrate all components onto the electrode surface. PANI provides a conductive matrix, AuNPs enhance electron transfer and surface area, GOx ensures glucose specificity, and BSA is used as a stabilizer to preserve enzyme activity. Although the base electrode is gold, the incorporation of AuNPs significantly enhances surface functionality and sensitivity. The resulting PANI-AuNPs-Gox-PPMP/SPE biosensor exhibits excellent analytical characteristics, including a low detection limit, rapid response time, strong selectivity against common interferents, and long-term operational stability. These features position it as a promising tool for cost-effective, portable, and accurate glucose monitoring in both clinical and industrial settings.

## 2. Experimental Section

### 2.1. Reagents and Solutions

Based on previously reported protocols [14,15,16,17], the following chemicals were used: sodium phosphate buffer (PB, pH 7.0), prepared using sodium dihydrogen phosphate (NaH_2_PO_4_, monobasic) and disodium hydrogen phosphate (Na_2_HPO_4_, dibasic), both purchased from Fisher Scientific (Waltham, MA, USA). Manganese (II) acetate tetrahydrate (Mn (CH_3_COO)_2_·4H_2_O, 9.99%), bovine serum albumin (BSA), glucose oxidase (Type X-S) from Aspergillus niger, and aniline were obtained from Sigma-Aldrich (St. Louis, MO, USA). Gold nanoparticles (GNPs) with an average diameter of 10 nm were also sourced from Sigma-Aldrich (St. Louis, MO, USA). All solutions were prepared using Milli-Q ultrapure water (resistivity: 18.2 MΩ·cm, MilliporeSigma, Burlington, MA, USA), and all experiments were conducted at room temperature under ambient laboratory conditions.

### 2.2. Apparatus

A computer controlled CHI660D electrochemical workstation (CH Instruments, Austin, TX, USA) was applied to perform all electrochemical measurements. Experiments were carried out using disposable/reusable screen-printed electrodes, including one 3 mm gold working electrode (WE), a silver reference electrode (RE), and a gold counter electrode (CE).

SPE|Polymer Matrix (PPMP (0.5 M)—Aniline (0.17 M)—Glucose Oxidase (0.10 M)—Bovine Serum Albumin (2.4 × 10^−6^ M)—Gold (Nanoparticles)| ×M Glucose, 0.1 M Phosphate-Buffer Solution (pH 7.0), 0.1 mM Manganese (II) Acetate Cell (1).

The composite mixture for electrode modification was prepared by combining the following components in 3.00 mL of phosphate-buffer solution (PB, pH ~7.0): PPMP (50.0 mg), aniline (50.0 mg), glucose oxidase (10.0 mg), bovine serum albumin (240 µL), and gold nanoparticles (0.50 g).

To prepare the 0.17 M aniline solution, 47.5 µL of aniline was first dissolved in 100 µL of absolute ethanol under gentle stirring. This ethanolic solution was then slowly added dropwise into 3.00 mL of PB while stirring to ensure homogeneity. After all components were thoroughly mixed, the resulting solution was sonicated for 10 min to achieve uniform dispersion.

The final composite mixture was applied to the surface of the screen-printed electrode (SPE) and subjected to electropolymerization. The modified electrodes were then tested with varying concentrations of glucose (×M) in 0.10 M PB (pH 7.0), with 0.10 mM manganese (II) acetate included in the electrolyte as an additional component.

Electrochemical measurements were carried out using a conventional three-electrode setup with a commercially available SPE (Model 220AT, Metrohm DropSens) as the sensing platform. These SPEs featured a gold working electrode, a gold counter electrode, and a silver reference electrode, all printed on a ceramic substrate using high-temperature ink. All experiments were performed in a 10 mL electrochemical cell at room temperature under static (non-stirred) conditions, unless otherwise specified. To ensure proper mixing, glucose solutions were briefly stirred with a small magnetic stirrer after each addition; however, all electrochemical measurements were performed under stationary conditions to maintain consistency and avoid signal interference caused by stirring. The working electrode surface was modified with the prepared composite, and measurements were conducted in the appropriate buffer solution across the selected glucose concentration range.

All electrochemical experiments were repeated at least three times independently. The results demonstrated good reproducibility, and error bars are shown in selected figures (Figures 2–9) to highlight analytical reproducibility; representative curves are presented in other figures for clarity.

### 2.3. PPMP Enzyme Preparation

Corn containing recombinant manganese peroxidase (MnP) was ground to a fine flour and extracted using 50 mM sodium tartrate buffer (pH 4.5; Fisher Scientific, BP352-500) at a ratio of 1.5 L buffer per 1 kg of corn. The pH of the sodium tartrate solution was adjusted to 4.5 with tartaric acid. Extraction was performed for 1 h in an ice bath under constant mixing.

A filtering aid (diatomaceous earth) was added to the resulting slurry, which was then filtered through a 15 cm Whatman #1 filter in a Büchner funnel under vacuum. The filtered solids were discarded, and the filtrate was concentrated to half of its original volume using a Pellicon-2 tangential flow filtration (TFF) ultrafiltration unit (Millipore). Solid ammonium sulfate was gradually added to achieve 95% saturation, and the mixture was stirred for approximately 1 h. Filtration aid was added, and the slurry was filtered using a Büchner funnel. The filtrate was discarded, and the precipitate was retained. The precipitate was resuspended in 50 mM sodium tartrate buffer (pH 4.5) at a concentration of 1 mg precipitate per 1 mL buffer and mixed for 30 min.

The slurry was again filtered (with filtration aid), retaining the filtrate and discarding the precipitate. The filtrate was concentrated and desalted using TFF (Pellicon-2, 10 kDa MW cutoff). This solution was passed over a Giga Cap S-650 M column to remove residual corn proteins, with the flow-through containing purified MnP. Fractions containing MnP were combined, concentrated via TFF, and lyophilized for storage.

### 2.4. Electrodeposition Procedure

An electrochemical method was employed to clean the SPEs prior to further modifications. Cyclic voltammetry was conducted in two distinct solutions: 1.0 M H_2_SO_4_ and Phosphate-Buffer Solution (PB).

For the cleaning process in 1.0 M H_2_SO_4_, the potential was swept between −0.1 V and +1.0 V for five cycles at a scan rate of 0.05 V/s. This cyclic voltammetry procedure effectively removes contaminants and impurities from the electrode surface, providing a clean starting point for subsequent modifications. The electrode was then cleaned in PBS under the same conditions to ensure the removal of any remaining unwanted substances, creating an optimal environment for future modifications. Following the cleaning steps, a 6 µL composite mixture containing PPMP, GOx, GNPs, and BSA was drop-cast onto the surface of the working electrode (WE). As illustrated in Figure 1 [14,15,16,17], the resulting glucose biosensor is based on a disposable screen-printed electrode platform and incorporates a recombinant corn-derived enzyme—PPMP, a redox-active peroxidase—for glucose detection. Despite the inclusion of multiple functional components, the fabrication process remains operationally straightforward. All modifiers are combined into a single solution This approach eliminates the need for sequential surface modifications or activation steps, significantly reducing fabrication time and complexity. Similar methods have been reported in the literature as scalable and cost-effective due to their compatibility with screen-printing and batch production techniques.

The biosensor functions through a multi-step enzymatic and electrochemical process involving glucose oxidase (GOx), the corn-derived peroxidase-like enzyme (PPMP), and bovine serum albumin (BSA), which facilitates enzyme stabilization and immobilization on the gold-modified electrode surface.

Glucose Oxidation by GOx catalyzes the oxidation of β-D-glucose to D-glucono-δ-lactone, accompanied by the reduction in molecular oxygen to hydrogen peroxide (H_2_O_2_):$$\mathrm{Glucose}+O_{2}\;\overset{{GO}_{X}}{\to }\;\mathrm{Gluconolactone}+H_{2}O_{2}$$

The recombinant peroxidase-like enzyme (PPMP), derived from corn (possibly related to manganese peroxidase, MnP), catalyzes the reduction in the generated H_2_O_2_ at the electrode surface. This reaction is critical for signal generation, producing a current proportional to the glucose concentration:$$H_{2}O_{2}+2H^{+}+2e\;\overset{PPMP}{\to }\;2H_{2}O$$

In this process, PPMP may utilize Mn^2+^ as a cofactor, oxidizing it to Mn^3+^, which further facilitates electron transfer and enhances signal amplification. BSA serves as a biocompatible stabilizing matrix, aiding in the co-immobilization of GOx and PPMP onto the gold-modified screen-printed electrode (SPE) surface. BSA helps maintain enzyme activity, prevents denaturation, and improves the mechanical stability of the modified electrode during storage and use in Figure 1.

The catalytic reaction of glucose oxidase (GOx) with glucose generates gluconolactone and hydrogen peroxide (H_2_O_2_), with molecular oxygen (O_2_) serving as the primary electron acceptor. In this coupled enzymatic system, oxygen plays two distinct roles. First, O_2_ is consumed during the GOx-catalyzed oxidation of glucose, producing gluconolactone and H_2_O_2_ as the main products. Second, during the subsequent PPMP peroxidase cycle, H_2_O_2_ undergoes catalytic turnover, where partial reduction steps may regenerate molecular oxygen. Thus, O_2_ functions both as a reactant in the initial oxidation and, under certain catalytic turnover conditions, as a byproduct of the PPMP-mediated redox process. The working electrode detects H_2_O_2_, which undergoes redox reactions at the electrode surface, producing an electrical signal proportional to the glucose concentration. This system employs a low-cost, single-use sensor strip, making it suitable for medical diagnostics and food industry applications. Following the drop-casting step, electropolymerization was performed on the SPEs. The cyclic voltammogram shown in Figure 1 depicts the electropolymerization process of the nanocomposite film on the electrode surface. The potential was cycled between −0.35 V and +0.45 V at a scan rate of 0.05 V/s, with the number of cycles varied from 15 to 30 to assess film growth and stability. Comparative analysis revealed that the film formed after 25 electropolymerization cycles demonstrated the most consistent electrochemical response and mechanical stability and was thus selected for further experiments. The observed redox peaks correspond to the oxidation and reduction in aniline monomers during the formation of the polyaniline film. The progressive increase in peak currents over successive cycles indicates the gradual growth of the electroactive polymer layer on the electrode surface. The electropolymerization solution in Cell 1 was prepared by dissolving PPMP, glucose oxidase, aniline (pre-dissolved in ethanol), bovine serum albumin (BSA), and gold nanoparticles in phosphate-buffer solution (PBS, pH 7.0), as described in Section 2.2. After modification, the electrodes were stored at 4 °C in an incubator when not in use to preserve their stability and ensure optimal performance.

## 3. Results and Discussion

### 3.1. Linear Sweep Voltammetry (LSV) Sensing of PANI-GNPs-GOx-PPMP/SPE

For the quantitative determination of glucose, linear sweep voltammetry (LSV) was employed. Measurements were performed in phosphate buffer (pH 7.0) containing 0.1 mM Mn (CH_3_COO)_2_ and saturated with oxygen. In LSV, the potential is swept linearly while recording the resulting current. Scans were conducted between 0.05 and 0.95 V at a scan rate of 0.05 V·s^−1^, with glucose added successively in the concentration range of 0.001–6.5 mM.

Figure 2A presents the background subtracted LSV curves, where the current response is expressed as ΔI (peak minus background). The increase in ΔI at ~0.58 V corresponds to the electrochemical oxidation of glucose. The anodic process at ~0.4 V can be [39,40] attributed primarily to the emeraldine/pernigraniline transition, while the overlapping irreversible feature may arise from cation-radical formation [41]. The lower-potential process is consistent with the leucoemeraldine/emeraldine couple. At this stage, the discussion focuses on the redox behavior of polyaniline; sensor performance metrics are discussed separately below.

The sensor exhibited high reproducibility across seven independently prepared electrodes, with error bars in Figure 2B representing the standard deviation of current responses. A strong linear correlation between current and glucose concentration was obtained (R^2^ = 0.9913), confirming both the reliability and sensitivity of the device. The limit of detection (LOD) was calculated as 0.50 µM using a signal-to-noise ratio of 3 and the formula LOD = (3 × standard deviation)/slope [42]. This value represents the minimum glucose concentration that can be reliably detected by the sensor.

### 3.2. Selectivity of the PANI-GNPs-GOx-PPMP/GSPE Using Linear Sweep Voltammetry (LSV)

To evaluate the selectivity of the developed glucose biosensor, potential interference from common electroactive species was investigated. At a concentration of 1.0 mM, caffeine—a compound frequently found in biological fluids and beverages—was selected as a representative interferent. For each interfering species, ΔI was measured in triplicate to ensure reproducibility. The sensor exhibited minimal interference, maintaining a clear and reliable glucose signal with a detection limit of 3.4 µM (R^2^ = 0.9953) (Figure 3A,B).

Similarly, aspartame and ascorbic acid were tested at the same concentration to assess their potential impact on glucose detection. The biosensor demonstrated low detection limits of 1.2 µM (R^2^ = 0.9955) for aspartame and 0.91 µM (R^2^ = 0.9944) for ascorbic acid, with negligible influence on the glucose signal. Although both caffeine and aspartame produced slight electrochemical responses, they did not significantly compromise glucose measurement accuracy. These findings highlight the biosensor’s strong selectivity and confirm its suitability for real-sample applications where such interferents may be present (Figure 4A,B and Figure 5A,B).

### 3.3. Square Wave Voltammetry (SWV) Sensing of PANI-GNPs-GOx-PPMP/GSPE

The SWV technique was employed to enhance sensitivity and minimize non-faradaic contributions, such as charging currents. SWV provided detailed insights into the electrochemical sensing behavior of the PANI-GNPs-GOx-PPMP/SPE modified electrode.

Electrodes were modified following the previously described protocol. SWV parameters were optimized to produce well-defined, peak-shaped voltammograms. The measurements were performed under the following conditions: pulse amplitude 25 mV, frequency 15 Hz, potential increment 4 mV, and sampling width 10 ms. Figure 6A shows the background subtracted SWV curves, with the Δ current used as the analytical signal. The potential was swept from 0.10 to 0.80 V, and glucose was incrementally added in the range of 0.0006 to 6.5 mM. A sharp increase in Δ current at approximately 0.58 V corresponds to the electrochemical oxidation of glucose.

Calibration curves were constructed by plotting the Δ current against glucose concentration (Figure 6B). The regression line exhibited excellent linearity (R^2^ = 0.998), indicating a strong correlation between glucose concentration and SWV response. The sensor achieved a limit of detection (LOD) of 0.29 µM, representing the lowest glucose concentration that could be reliably detected.

### 3.4. Selectivity of the PANI-GNPs-GOx-PPMP/GSPE Using Square Wave Voltammetry (SWV)

For each interfering species, ΔI was measured in triplicate to ensure reproducibility. Figure 7A,B present square wave voltammetry (SWV) results assessing the selectivity of the developed glucose biosensor in the presence of caffeine, a common electroactive compound that may interfere with glucose detection. The biosensor exhibited a well-defined and distinct glucose signal even in the presence of 1.0 mM caffeine, with only a minimal shift in the current response. This indicates that caffeine has a negligible effect on sensor performance, which maintained a detection limit of 0.55 µM (R^2^ = 0.993). These results support the biosensor’s capability to accurately detect glucose in complex matrices such as beverages and biological fluids.

Figure 8A,B further evaluate selectivity by examining the sensor’s response to 1.0 mM aspartame, a widely used artificial sweetener often coexisting with glucose in food samples. The biosensor again produced a clear glucose signal with minimal interference, achieving a slightly improved detection limit of 0.40 µM (R^2^ = 0.9917). These findings highlight the sensor’s high selectivity and suitability for food-related applications.

Figure 9A,B show the sensor’s performance in the presence of 1.0 mM ascorbic acid, a common antioxidant found in both biological fluids and food products. The biosensor maintained a distinct glucose signal with negligible interference, achieving a detection limit of 0.44 µM (R^2^ = 0.9951).

Overall, the consistent performance across all tested interferents demonstrates the biosensor’s strong selectivity, sensitivity, and potential for reliable glucose monitoring in real-world applications, including food quality control and point-of-care diagnostics.

### 3.5. Comparison of Electrochemical Glucose Sensors: Advantages of a Novel Electropolymerized SWV-Based Platform

Electrochemical glucose sensors commonly employ amperometric or chronoamperometric techniques due to their simplicity, high sensitivity, and rapid response times. Traditional amperometric sensors operate by applying a constant potential and measuring the resulting current generated from the enzymatic oxidation of glucose—typically facilitated by glucose oxidase (GOx) immobilized on the electrode surface—with hydrogen peroxide produced as a measurable byproduct (Table 1) [43,44,45,46,47,48,49]. In comparison to the sensors listed in Table 1, our biosensor exhibits significantly enhanced analytical performance, particularly in terms of detection limit and linear range. By employing both square wave voltammetry (SWV) and linear sweep voltammetry (LSV), the sensor achieved exceptionally low detection limits of 0.29 µM (SWV) and 0.29 µM (LSV), outperforming many previously reported screen-printed electrode (SPE)-based biosensors. Moreover, the biosensor provides an extended linear detection range of 0.0006 to 6.5 mM, which exceeds those typically observed in similar devices. This enhanced performance is attributed to the integration of a recombinant plant-derived enzyme (PPMP), glucose oxidase, and gold nanoparticles on a gold-modified SPE. Selectivity testing in the presence of caffeine confirmed the sensor’s reliability in complex sample environments. The combination of low fabrication cost, high sensitivity, and wide applicability underscores the biosensor’s potential for real-world use in point-of-care diagnostics and food quality monitoring.

### 3.6. Comparative Analysis with Previous Works

Our previous studies Table 2 have demonstrated the enzymatic activity and effectiveness of the recombinant corn-derived enzyme, PPMP, in enhancing the performance of electrochemical biosensors. Specifically, our research has consistently shown the efficacy of PPMP in developing sensitive and selective biosensors for the detection of glucose and hydrogen peroxide (Table 2).

In our initial study, we developed a Nafion/PPMP-Gox-BSA/Au biosensor that exhibited excellent amperometric performance across a broad glucose concentration range (20.0 μM to 15.0 mM), with a low detection limit of 2.9 μM. The biosensor also demonstrated strong selectivity in complex matrices such as diet green tea. Building on this success, we later employed PPMP to fabricate a novel biosensor for hydrogen peroxide detection. This sensor achieved a wide linear range of 0.005–2.5 mM and a detection limit as low as 0.29 μM, demonstrating robust performance in both food and environmental sample analyses.

Our third investigation transitioned from traditional electrodes to a miniaturized 10 µm diameter gold microelectrode (GME), combining polyaniline (PANI), gold nanoparticles (GNPs), GOx, and PPMP to achieve a remarkably low glucose detection limit of 0.5 µM and a broad linear range (0.001–16.0 mM). Finally, we further optimized the biosensor matrix by polymerizing aniline in the presence of AuNPs-GOx-PPMP and BSA, achieving an extended linear detection range (0.005–16.0 mM) and an even lower detection limit of 0.001 mM using both LSV and CV techniques.

Table 3 Shows glucose concentrations vary markedly across different biofluids—ranging from 3.3 to 17.3 mM in blood, 0.1 to 0.6 mM in tears, and 0.02 to 0.6 mM in sweat—necessitating biosensors with wide dynamic ranges and high sensitivity to enable reliable detection in both physiological and trace-level conditions. In this context, our work offering rapid response, ease of use, and minimal risk of cross-contamination, the biosensor is promising for point-of-care diagnostics and glucose monitoring in clinical and food applications. This work underscores the potential of bioengineered plant enzymes integrated with nanomaterial-enhanced electrodes for developing the next generation of scalable, low-cost, and high-performance biosensors.

In the current work, we advance this platform by integrating gold modified screen-printed electrodes (SPEs) with PPMP and GOx and applying both LSV and Square Wave Voltammetry (SWV). This novel configuration not only offers improved miniaturization and cost-efficiency but also achieves a superior limit of detection (LOD = 0.29 µM with SWV), surpassing the sensitivity of our earlier PPMP-based glucose biosensors. Additionally, the biosensor exhibits enhanced selectivity against common interferents such as ascorbic acid, dopamine, uric acid, and aspartame, demonstrating strong potential for point-of-care and food monitoring applications. These results highlight the versatility and continued promise of plant-derived enzymes in developing next-generation biosensing platforms.

## 4. Conclusions

In this study, we developed a novel, disposable electrochemical biosensor for glucose detection by integrating a recombinant corn-derived enzyme (PPMP) with glucose oxidase (GOx), gold nanoparticles (GNPs), bovine serum albumin (BSA), and gold-modified screen-printed electrodes (GSPEs). Glucose detection by LSV covered a linear range of 0.001–6.5 mM (LOD = 0.50 µM), while SWV achieved an even broader range of 0.0006–6.5 mM (LOD = 0.29 µM), surpassing previously reported PPMP-based sensors in sensitivity and detection limits.

Compared to earlier designs employing conventional gold electrodes and microelectrodes, this biosensor offers marked advantages in sensitivity, ease of fabrication, and scalability. It demonstrated excellent selectivity, showing minimal interference from common electroactive species—including ascorbic acid, caffeine, and aspartame—while maintaining clear, reliable glucose signals even in the presence of 1.0 mM interferents, confirming its suitability for real-sample analysis.

The fabrication method—a single-step drop-casting of a pre-mixed modifier solution onto SPEs—ensures a simple, reproducible, and time-efficient process. Combined with the compatibility of SPEs with scalable manufacturing techniques such as roll-to-roll or stencil printing, this biosensor design is ideally suited for miniaturization and cost-effective mass production. These features highlight its strong potential for broad deployment in clinical diagnostics, food quality control, and portable point-of-care applications.

## Data Availability

All data generated or analyzed during this study are included in this published article.

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
