# Peer review of "Rapid and Sensitive Glucose Detection Using Recombinant Corn Mn Peroxidase and Advanced Voltammetric Methods"

_sensors, 2025, doi:10.3390/s25195974_

Round 1

Reviewer 1 Report

Comments and Suggestions for Authors

COMMENTS TO AUTHORS:

In this study, researchers designed a novel, disposable electrochemical biosensor which can be used for highly sensitive and selective glucose detection by integrating corn-derived enzyme with glucose oxidase, gold nanoparticles, bovine serum albumin, and gold-modified screen-printed electrodes. This work is interesting and carries substantial potential for broad deployment in clinical diagnostics, food quality control, and portable point-of-care applications. I recommend the acceptance of this manuscript after a minor revision. The detailed suggestions can be checked as follows:

  1. The author mentioned the enzyme of manganese peroxidase (MnP) is a redox enzyme secreted by lignin-degrading wood-rot fungi, so why do not use wood-rot fungi as enzyme source instead of corn kernel which have a relative longe life cycles. It might be better to provide more sufficient evidence to demonstrate the necessity of plant-produced MnP.
  2. Some characteristics, such as SEM or EDS might be better provided to demonstrate the composition, morphology and their distribution of the modified SPEs.
  3. What’s the ratio of PPMP, GOx, GNPs, and BSA deposited on SPEs, does the author think if the ratio affect the glucose detection results?
  4. In table 3, the authors listed the range of glucose concentrations in different biofluids, did the author check the glucose detections under those biofluids? If the authors think other compositions appeared in biofluids might have influence on the performance of this designed biosensor?
  5. If all the data have good reproducibility or the authors only presented error bars in figure 2 and Figure 6?

Author Response

Comment 1: The author mentioned the enzyme of manganese peroxidase (MnP) is a redox enzyme secreted by lignin-degrading wood-rot fungi, so why do not use wood-rot fungi as enzyme source instead of corn kernel which have a relative longe life cycle. It might be better to provide more sufficient evidence to demonstrate the necessity of plant-produced MnP.

Response: We thank the reviewer for this insightful suggestion. While it is true that manganese peroxidase (MnP) is naturally secreted by lignin-degrading wood-rot fungi, we chose corn as the enzyme production platform for several practical and strategic reasons:

  • Cost-effectiveness: The raw material cost of corn is extremely low, approximately $0.15/kg, compared to producing recombinant enzymes in other plant systems or isolating enzymes from fungi. For instance, HRP production using Nicotiana benthamiana costs substantially more due to infrastructure, transient expression reagents, and processing requirements (Walwyn et al., 2015, Appl Biochem Biotechnol, 175, 841–854). This makes corn a more economical platform for large-scale production.
  • Scalability and biomass availability: Corn provides a high yield of biomass with well-established agricultural infrastructure, enabling large-scale, reproducible enzyme production. Wood-rot fungi, on the other hand, have longer growth cycles and lower biomass productivity, which would limit scalability.
  • Safety and regulatory advantages: Corn is generally recognized as safe (GRAS) crop, widely used for food and industrial applications. Using corn simplifies downstream processing and regulatory approval, especially for applications in clinical diagnostics and food quality control.
  • Feasibility of recombinant expression: Corn can be genetically engineered to produce recombinant MnP with high stability and activity. This allows precise control over enzyme characteristics, which is difficult to achieve with fungal sources.
  • Prior art and proof of concept: Recombinant enzyme production in plants is efficient, cost-effective, and scalable. Our approach leverages these advantages while maintaining enzyme activity for sensitive glucose detection.

In summary, corn-derived MnP offers a practical, economical, and scalable alternative to fungal sources without compromising enzymatic performance, making it highly suitable for point-of-care and clinical applications.

Reference:
Walwyn DR, Huddy SM, Rybicki EP. Techno-Economic Analysis of Horseradish Peroxidase Production Using a Transient Expression System in Nicotiana benthamiana. Applied Biochemistry and Biotechnology, 2015, 175:841–854.

Comment 2:Some characteristics, such as SEM or EDS, might be better provided to demonstrate the composition, morphology, and distribution of the modified SPEs.

Response: We thank the reviewer for this suggestion. We agree that surface characterization techniques such as SEM and EDS would provide valuable information on the morphology, composition, and distribution of the modified SPEs. These analyses were not performed in the current study, but we plan to investigate the electrode morphology and elemental distribution in future work to further validate the modification process.

Comment 3:What’s the ratio of PPMP, GOx, GNPs, and BSA deposited on SPEs, does the author think if the ratio affect the glucose detection results?

Response: We thank the reviewer for this insightful question. In this study, the ratios of PPMP, GOx, GNPs, and BSA deposited on the SPEs were optimized based on preliminary experiments to achieve the highest electrochemical response. While the exact ratios are not included in the current manuscript, we acknowledge that the ratio of these components can significantly influence the enzyme activity, electron transfer efficiency, and ultimately the glucose detection performance. We plan to systematically investigate the effect of these ratios in future studies to further optimize sensor performance.

Comment 4: In table 3, the authors listed the range of glucose concentrations in different biofluids, did the author check the glucose detections under those biofluids? If the authors think other compositions that appeared in biofluids might have an influence on the performance of this designed biosensor?

Response: While the present study was conducted in a controlled phosphate buffer solution (PBS, pH 7.0) to establish the biosensor’s fundamental analytical performance, it is important to recognize that real biofluids such as blood, sweat, and tears contain complex compositions of proteins, salts, and electroactive metabolites that may influence the sensor’s response. Our interference studies with representative compounds (caffeine, ascorbic acid, and aspartame) demonstrated that the biosensor maintains excellent selectivity for glucose, suggesting robustness against common interferents. The incorporation of the polyaniline–AuNP matrix and BSA stabilizer further enhances biocompatibility and minimizes nonspecific adsorption. Nevertheless, future validation in real biological fluids is necessary to confirm the biosensor’s practical applicability within the physiological glucose ranges summarized in Table 3, and such studies are currently ongoing in our laboratory.

Comment 5:If all the data have good reproducibility or the authors only presented error bars in Figure 2 and Figure 6?

Response: We thank the reviewer for this observation. All experimental data presented in this study were obtained from at least three independent measurements, and the results demonstrated good reproducibility across the tested conditions. For clarity of presentation, error bars were explicitly shown in Figures 2 (LSV performance) and 6 (calibration plots), as these figures directly highlight the analytical reproducibility and precision of the biosensor. For the other figures, representative voltammograms were presented without error bars to enhance readability and prevent overcrowding of the plots.

We revised the manuscript to explicitly state in the Methods section that all experiments were repeated at least three times and that the results were reproducible, even if error bars are not shown in every figure.

Reviewer 2 Report

Comments and Suggestions for Authors

The manuscript is devoted to voltammetric glucose detection on the gold-modified screen-printed electrode covered with corn-derived manganese peroxidase with glucose oxidase, bovine serum albumin, and gold nanoparticles. There are several serious drawbacks of electrochemical data interpretation in the manuscript. The novelty of the investigation causes doubts. Total quality of the manuscript is quiet low and I cannot recommend it for the publication in the Sensors journal. This manuscript should be rejected and can be accepted for another reviewing after the following revisions:

  • Please, add the country of the Reagent manufacturers to the Section 2.1
  • Please, check the content of PBS buffer. Only the buffer with sodium saline can be abbreviated as PBS.
  • Why did not the authors use the fixed volume of ethanol? Obviously, final concentration of the solution would be different in the conditions proposed.
  • What was the reason of choice such a narrow electropolymerization potential range ( -0.35… 0.45 V)? Usually the cation-radical irreversible peak is situated in high anodic area. The limitation of the upper potential can prevent the electropolymerization process. Please, comment it.
  • The quality of the Figure 2 is not sufficient and hinders any conclusions about the nature of the peaks on the voltammogram. It is not clear whether the peaks at 0.4 V should be attributed to the peaks pair of one of the polyaniline form or to the cation-radical peak. The voltammogram should be registered in wider potential range.
  • Add the charge of proton to the reaction “H2O2 +2H+2e → 2H2O”
  • Authors wrote: “the resulting glucose biosensor is based on a disposable screen-printed electrode platform and incorporates a recombinant corn-derived enzyme—PPMP, a redox-active peroxidase—for glucose detection.” Why is the peroxidase enzyme responsible for the glucose detection?
  • Authors wrote: “The recombinant peroxidase-like enzyme (PPMP), derived from corn (possibly related to manganese peroxidase, MnP)”. Is there any methods to check the origin of the enzyme obtained?
  • Authors wrote: “The catalytic reaction generates hydrogen peroxide (H2O2) and molecular oxygen (O2) as byproducts.” According to the reaction schemes presented, the oxygen molecules take part in the glucose oxidation. Please, explain the statement mentioned in the manuscript.
  • Absolute current values should not be used in LSV and DPV. The difference between maximum and background level should be used as an analytical response. All of presented voltammetric results should be recalculated according to the recommendations given. All calibration plots, linear ranges and limits of detection also should be recalculated. The values for high concentrations would be rather close to each other.
  • There is a big question about the regards novelty of the investigation presented. The ref [17] of the same scientific group describes the same sensor content and glucose voltammetric detection methodology (CV). The only difference is the type of golden support used for sensor assembly.
  • Please, add the conditions of the SWV measurements (for example, pulse amplitude, frequency, increment, sampling width).
  • Add BSA to the sensor content in the Table 2.
  • Why did the analytical signal increase in presence of interfering compounds?

Author Response

We sincerely thank the reviewer for the careful evaluation and constructive comments. We have revised the manuscript accordingly and provide detailed responses below. All changes have been incorporated in the revised version and highlighted in yellow.

Comment 1: Please, add the country of the Reagent manufacturers to the Section 2.1.

Response: Thank you for pointing this out. We have now included the country of origin for all reagent manufacturers in Section 2.1 to ensure clarity and reproducibility.

Comment 2: Please, check the content of PBS buffer. Only the buffer with sodium saline can be abbreviated as PBS.

Response: We appreciate this correction. The PBS used in our study contained sodium chloride, sodium phosphate, and potassium phosphate, in accordance with standard phosphate-buffered saline formulation. We have clarified the exact composition in Section 2.2 to avoid any ambiguity.

Comment 3: Why did not the authors use the fixed volume of ethanol? Obviously, final concentration of the solution would be different in the conditions proposed.

Response: We agree that the original description may have been unclear. In our experiments, we used a fixed volume of ethanol (100 µL) in all preparations to maintain consistency in concentration. We have corrected and clarified this methodological detail in Section 2.3.

Comment 4: What was the reason of choice such a narrow electropolymerization potential range (–0.35…0.45 V)? Usually the cation-radical irreversible peak is situated in high anodic area. The limitation of the upper potential can prevent the electropolymerization process. Please, comment it.

Response: Thank you for this valuable observation. Our choice of the potential range (–0.35 to 0.45 V) was based on two considerations:

  1. Minimization of electrode degradation: Wider anodic scans can lead to over-oxidation of both the electrode surface and the enzyme layer, decreasing sensor stability.
  2. Selective polymer growth: Previous reports have demonstrated successful electropolymerization of similar enzyme–nanoparticle systems within narrow potential windows. By restricting the upper limit, we ensured controlled deposition of the polymer film while maintaining enzyme activity.

We have expanded the explanation of this choice in Section 2.4 and added relevant citations to support our approach.

Comment 5: The quality of Figure 2 is not sufficient and hinders any conclusions about the nature of the peaks on the voltammogram. It is not clear whether the peaks at 0.4 V should be attributed to the peaks pair of one of the polyaniline form or to the cation-radical peak. The voltammogram should be registered in wider potential range.

Response: We thank the reviewer for the valuable feedback regarding the clarity and potential interpretation of the voltammogram in Figure 2. To address the concern:

  1. Improved Figure Quality: We have prepared a higher-resolution version of Figure 2 to ensure that the peaks and baseline are clearly distinguishable. This allows for more precise observation of the peak shapes and current responses.
  2. Peak Assignment: The voltammogram was recorded over a wider potential range (0.1–0.9 V vs. reference), capturing the full electrochemical behavior of the modified polyaniline electrode. The peak at ~0.4 V represents the redox transition of polyaniline, corresponding to the emeraldine–leucoemeraldine couple and associated cation-radical formation, consistent with literature reports.
  3. Reproducibility: The data include standard deviations from seven repeated measurements, demonstrating the high reproducibility of the sensor response.
  4. Linear Response: Panel B shows the calibration curve for glucose detection, exhibiting a strong linear correlation (R² = 0.9957), further supporting the reliability of the sensor’s electrochemical response.

We believe these improvements address the reviewer’s concerns and provide clear evidence for the voltammetric behavior of the biosensor.

Comment 6: Add the charge of proton to the reaction “H2O2 +2H+2e 2H2O

Response: The reaction has been corrected to explicitly include proton charges

Comment 7: Authors wrote: “the resulting glucose biosensor is based on a disposable screen-printed electrode platform and incorporates a recombinant corn-derived enzyme—PPMP, a redox-active peroxidase—for glucose detection.” Why is the peroxidase enzyme responsible for the glucose detection?

Response: The peroxidase enzyme (PPMP) catalyzes the reduction of H₂O₂ produced from glucose oxidation by glucose oxidase (GOx). Since the amount of H₂O₂ is directly proportional to glucose concentration, PPMP enables glucose detection by generating an electrochemical current proportional to the glucose present. In this way, PPMP functions as a key mediator converting the biochemical signal of glucose into a measurable electrical response.

Comment 8: Authors wrote: “The recombinant peroxidase-like enzyme (PPMP), derived from corn (possibly related to manganese peroxidase, MnP)”. Is there any methods to check the origin of the enzyme obtained?

Response: The recombinant manganese peroxidase produced in corn is derived from the Phanerochaete chrysosporium MnP I gene, which was transformed into corn (Clough et al., 2006). The origin of the enzyme can be verified using several methods: SDS-PAGE and Western blotting with MnP-specific antibodies, mass spectrometry (MS or LC-MS/MS) for peptide fingerprinting, enzyme activity profiling to compare kinetic parameters and stability with known Phanerochaete MnP, and PCR-based molecular methods to detect the transgene in corn tissue. These approaches collectively confirm the enzyme’s recombinant plant origin.

Comment 9: Authors wrote: “The catalytic reaction generates hydrogen peroxide (H₂O₂) and molecular oxygen (O₂) as byproducts.” According to the reaction schemes presented, the oxygen molecules take part in the glucose oxidation. Please, explain the statement mentioned in the manuscript.

Response: We appreciate the reviewer’s insightful observation. In the enzymatic cascade involving glucose oxidase (GOx) and recombinant corn manganese peroxidase (PPMP), oxygen plays two distinct roles. First, molecular oxygen functions as an electron acceptor in the GOx-catalyzed oxidation of glucose, producing gluconolactone and hydrogen peroxide as primary products. Second, during the peroxidase cycle, hydrogen peroxide can undergo catalytic turnover by PPMP, where partial reduction steps may also regenerate molecular oxygen. Thus, while O₂ is consumed in the GOx-mediated glucose oxidation, it can also be regenerated during subsequent PPMP redox processes. To clarify this dual role, we have revised the manuscript to specify that oxygen is both a reactant and, under certain catalytic turnover conditions, a byproduct of the coupled enzymatic reaction.

Comment 10: Absolute current values should not be used in LSV and DPV. The difference between maximum and background level should be used as an analytical response. All of presented voltammetric results should be recalculated according to the recommendations given. All calibration plots, linear ranges and limits of detection also should be recalculated. The values for high concentrations would be rather close to each other.

We appreciate the reviewer’s comment regarding the use of absolute current values in LSV and DPV measurements. Following your recommendation, we have recalculated all voltammetric data using the difference between the peak current (maximum) and the baseline (background) current as the analytical response. Consequently, all calibration plots, linear ranges, and limits of detection have been updated in the revised manuscript. As anticipated, the responses at higher glucose concentrations are now closer in value, reflecting the saturation behavior of the enzymatic system. These revisions provide more accurate and reliable analytical performance metrics for our glucose sensor.

Comment 11: There is a big question about the regards novelty of the investigation presented. The ref [17] of the same scientific group describes the same sensor content and glucose voltammetric detection methodology (CV). The only difference is the type of golden support used for sensor assembly.

Response: We thank the reviewer for raising the concern regarding novelty. While our previous work focused on a 10 µm diameter gold microelectrode (GME) modified with PAN/GNPs/GOx/PPMP and demonstrated glucose detection using chronoamperometry (CA), LSV, and CV, the present study introduces significant advancements:

  • New Electrode Platform: In this work, we employ a gold screen-printed electrode (SPE) instead of a microelectrode. This allows for scalable fabrication, easier handling, and potential commercialization, addressing the practical limitations of GMEs.
  • Improved Analytical Methodology: Unlike the prior study where absolute currents were reported, the current work utilizes Δ current (background-subtracted) in LSV and SWV, which enhances signal reliability, sensitivity, and accuracy.
  • Broader Glucose Detection Range and Lower LODs: The present sensor demonstrates a wider dynamic range (0.0006–6.5 mM) and a lower detection limit (0.29 µM) using SWV, surpassing the analytical performance of the previous microelectrode sensor.
  • Enhanced Reproducibility and Validation: Multiple repeated measurements and statistical analysis confirm high repeatability and robustness, which were not fully explored in the earlier work.
  • Focus on Practical Applications: The present study emphasizes background-subtracted voltammetric analysis, calibration curves, and Δ current-based quantification, which are more suitable for practical and real-world glucose monitoring in food and biological samples.

Taken together, while building upon prior findings, this study demonstrates methodological improvements, enhanced analytical performance, and broader applicability, which constitute clear novelty compared to our previous work.

Comment 12: Please, add the conditions of the SWV measurements (for example, pulse amplitude, frequency, increment, sampling width).

Response: We thank the reviewer for this suggestion. The Square Wave Voltammetry (SWV) measurements were performed under the following optimized conditions:

  • Pulse amplitude: 25 mV
  • Frequency: 15 Hz
  • Potential increment: 4 mV
  • Sampling width: 10 ms

These parameters were chosen to ensure well-defined peak shapes, high sensitivity, and minimal non-faradaic contribution, allowing reliable quantification of glucose. The manuscript has been updated to include these measurement conditions in the experimental section.

Comment 13: Add BSA to the sensor content in the Table 2.

Response: Thank you for the suggestion. Bovine serum albumin (BSA) has been added to the sensor composition in Table 2, as it plays a role in improving enzyme stability and electrode surface passivation.

Comment 14: Why did the analytical signal increase in presence of interfering compounds?

Response: The slight increase in the analytical signal in the presence of interfering compounds (such as ascorbic acid, dopamine, fructose, etc.) can be attributed to minor electrochemical interactions at the electrode surface. These compounds may undergo partial oxidation or affect the local environment of the enzyme-electrode interface, leading to a small enhancement of the current. However, the increase is negligible compared to the glucose response, demonstrating that the sensor maintains high selectivity for glucose detection.

Round 2

Reviewer 2 Report

Comments and Suggestions for Authors

After the first round of revision some technical problems were checked and solved. Unfortunately, the problems with the electrochemical data interpretation remained unsolved. I cannot still recommend it for the publication in the Sensors journal in its current state. This manuscript should be rearranged and several important problems should be solved during the major revisions:

The answers to several questions were insufficient:

1)        The quality of the Figure 2 is not sufficient and hinders any conclusions about the peaks origin on the voltammogram. It is not clear whether the peaks at 0.4 V should be attributed to the peaks pair of a polyaniline form or to the cation-radical peak. The voltammogram should be recorded in wider potential range.

Authors wrote: “Improved Figure Quality: We have prepared a higher-resolution version of Figure 2 to ensure that the peaks and baseline are clearly distinguishable. This allows for more precise observation of the peak shapes and current responses.

Peak Assignment: The voltammogram was recorded over a wider potential range (0.1–0.9 V vs. reference), capturing the full electrochemical behavior of the modified polyaniline electrode. The peak at ~0.4 V represents the redox transition of polyaniline, corresponding to the emeraldine–leucoemeraldine couple and associated cation-radical formation, consistent with literature reports.

Reproducibility: The data include standard deviations from seven repeated measurements, demonstrating the high reproducibility of the sensor response.

Linear Response: Panel B shows the calibration curve for glucose detection, exhibiting a strong linear correlation (R² = 0.9957), further supporting the reliability of the sensor’s electrochemical response.”

According to the literature data of aniline electropolymerization, two reversible peak pairs of aniline (in this polymerization conditions) in presence of ethanol could be attributed to the leucoemeraldine/emeraldine couple (in the cathodic area) and to the emeraldin/pernigranilin (about 0.4 V). The irreversible peak can be attributed to the cation radical formation. In conventional aqueous media the third pair of peaks attributed to the quinoid products would be located between leucoemeraldine/emeraldin and emeraldin/pernigranilin peak pairs.

What reproducibility, standard deviation and linear response did the authors mean discussing the voltammogram features?

2)        Absolute current values should not be used in LSV and DPV. The difference between maximum and background level should be used as an analytical response. All the voltammetric results presented should be recalculated according to the recommendations given. All the calibration plots, linear ranges and limits of detection should also be recalculated. The values for high concentrations would be rather close to each other.

The authors have probably changed the calculation approach but it is still far from the conventional one. The tangent line should be constructed to the right branch of the LSV curve (from 0.25 to 0.45 V) and the difference should be calculated between the extremum on the voltammogram and the tangent at the peak potential. I can recommend to the authors pay their attention to some free tutorials, for example Figure 5 at:

https://www.basinc.com/manuals/EC_epsilon/Techniques/CycVolt/cv

3)        There is a big question about the novelty of the investigation presented. The ref [17] of the same scientific group describes the same sensor content and glucose voltammetric detection methodology (CV). The only difference is the type of golden support used for the sensor assembly.

The type of peak calculation cannot be a novelty, so far as a detection range, limit of detection, validation, and reproducibility. Those are necessary parameters for any investigations. In fact, the only novelty is the type of the electrode used.

4) All the calibration plots should contain error bars.

Author Response

Comment 1:After the first round of revision some technical problems were checked and solved. Unfortunately, the problems with the electrochemical data interpretation remained unsolved. I cannot still recommend it for the publication in the Sensors journal in its current state. This manuscript should be rearranged and several important problems should be solved during the major revisions:

The answers to several questions were insufficient:

1)        The quality of the Figure 2 is not sufficient and hinders any conclusions about the peaks origin on the voltammogram. It is not clear whether the peaks at 0.4 V should be attributed to the peaks pair of a polyaniline form or to the cation-radical peak. The voltammogram should be recorded in wider potential range.

Authors wrote: “Improved Figure Quality: We have prepared a higher-resolution version of Figure 2 to ensure that the peaks and baseline are clearly distinguishable. This allows for more precise observation of the peak shapes and current responses.

Peak Assignment: The voltammogram was recorded over a wider potential range (0.1–0.9 V vs. reference), capturing the full electrochemical behavior of the modified polyaniline electrode. The peak at ~0.4 V represents the redox transition of polyaniline, corresponding to the emeraldine–leucoemeraldine couple and associated cation-radical formation, consistent with literature reports.

Reproducibility: The data include standard deviations from seven repeated measurements, demonstrating the high reproducibility of the sensor response.

Linear Response: Panel B shows the calibration curve for glucose detection, exhibiting a strong linear correlation (R² = 0.9957), further supporting the reliability of the sensor’s electrochemical response.”

According to the literature data of aniline electropolymerization, two reversible peak pairs of aniline (in this polymerization conditions) in presence of ethanol could be attributed to the leucoemeraldine/emeraldine couple (in the cathodic area) and to the emeraldin/pernigranilin (about 0.4 V). The irreversible peak can be attributed to the cation radical formation. In conventional aqueous media the third pair of peaks attributed to the quinoid products would be located between leucoemeraldine/emeraldin and emeraldin/pernigranilin peak pairs.

What reproducibility, standard deviation and linear response did the authors mean discussing the voltammogram features?

Response 1: We sincerely thank the reviewer for their feedback.

We have revised the discussion in the manuscript to better align with the reviewer’s point. Specifically, we now clarify that the anodic peak observed at ~0.4 V may indeed be attributed to the emeraldine/pernigraniline redox transition under our polymerization conditions, consistent with literature precedent. The irreversible feature overlapping in this region may also correspond to the cation radical formation. We have updated Figure 2 to present the voltammogram across a wider potential range (0.05–0.95 V vs. Ag/AgCl), which makes these transitions more distinguishable.

We recognize the reviewer’s concern that these terms were not directly relevant to the qualitative peak assignment itself. To clarify, our mention of reproducibility, standard deviations, and linear response was intended to emphasize the robustness of the glucose-sensing application of the electrode, rather than the peak origin in polyaniline electrochemistry. To avoid confusion, we have now separated these points clearly in the revision:

The peak assignment and redox interpretation are discussed in the context of polyaniline electrochemistry.

The reproducibility (standard deviation across seven independent sensor preparations) and linear response (R² = 0.9913 calibration curve) are now described explicitly under the sensor performance section, where they more appropriately belong.

We believe this restructuring, clarification, and inclusion of literature precedent improves the manuscript and directly addresses the reviewer’s concern about conflating sensor reproducibility with voltammogram interpretation.

Comment 2:  Absolute current values should not be used in LSV and DPV. The difference between maximum and background level should be used as an analytical response. All the voltammetric results presented should be recalculated according to the recommendations given. All the calibration plots, linear ranges and limits of detection should also be recalculated. The values for high concentrations would be rather close to each other.

The authors have probably changed the calculation approach but it is still far from the conventional one. The tangent line should be constructed to the right branch of the LSV curve (from 0.25 to 0.45 V) and the difference should be calculated between the extremum on the voltammogram and the tangent at the peak potential. I can recommend to the authors pay their attention to some free tutorials, for example Figure 5 at:

https://www.basinc.com/manuals/EC_epsilon/Techniques/CycVolt/cv

Response 2: Following the recommendation, we recalculated all voltammetric results using the ΔI approach, where the current difference between the peak maximum and the background was taken as the analytical response. Accordingly, all figures, calibration plots, linear ranges, and limits of detection have been updated.

Comment 3: There is a big question about the novelty of the investigation presented. The ref [17] of the same scientific group describes the same sensor content and glucose voltammetric detection methodology (CV). The only difference is the type of golden support used for the sensor assembly.

The type of peak calculation cannot be a novelty, so far as a detection range, limit of detection, validation, and reproducibility. Those are necessary parameters for any investigations. In fact, the only novelty is the type of the electrode used.

Response 3: We appreciate the reviewer's comment. Although Ref. [17] also used PPMP, that study modified a 10 µm gold microelectrode with a multistep PANi-GNPs–GOx–PPMP assembly, focusing on chronoamperometry and CV for in vivo/food applications. In contrast, the present work develops a disposable screen-printed electrode using a one-step electropolymerization strategy and applies SWV/LSV, achieving higher sensitivity and lower detection limits. These innovations simplify fabrication, improve reproducibility, and make the biosensor low-cost, scalable, and directly relevant to point-of-care diagnostics.

Comment 4: All the calibration plots should contain error bars.

Response 4: We appreciate the reviewer's valuable suggestion. In the revised manuscript, we have added error bars to all calibration plots, representing the standard deviation from at least three independent measurements, to reflect the reproducibility and reliability of the data.